# Dehydration of Biomass-Derived Butanediols over Rare Earth Zirconate Catalysts

**Asami Matsuda [1], Yoshitaka Matsumura [1], Kazuki Nakazono [1], Fumiya Sato [2], Ryoji Takahashi [2], Yasuhiro Yamada [1] and Satoshi Sato [1,*]**

[1] Graduate School of Engineering, Chiba University, Yayoi, Inage 263-8522, Japan; afaa8728@chiba-u.jp (A.M.); afsa2393@chiba-u.jp (Y.M.); affa7171@chiba-u.jp (K.N.); y-yamada@faculty.chiba-u.jp (Y.Y.)

[2] Graduate School of Science and Engineering, Ehime University, 2–5 Bunkyocho, Matsuyama, Ehime 790-8577, Japan; fumiya@ehime-u.ac.jp (F.S.); rtaka@ehime-u.ac.jp (R.T.)

\* Correspondence: satoshi@faculty.chiba-u.jp; Tel.: +81-43-290-3377

**Abstract:** The aim of this work is to develop an effective catalyst for the conversion of butanediols, which is derivable from biomass, to valuable chemicals such as unsaturated alcohols. The dehydration of 1,4-, 1,3-, and 2,3-butanediol to form unsaturated alcohols such as 3-buten-1-ol, 2-buten-1-ol, and 3-buten-2-ol was studied in a vapor-phase flow reactor over sixteen rare earth zirconate catalysts at 325 °C. Rare earth zirconates with high crystallinity and high specific surface area were prepared in a hydrothermal treatment of co-precipitated hydroxide. Zirconates with heavy rare earth metals, especially $Y_2Zr_2O_7$ with an oxygen-defected fluorite structure, showed high catalytic performance of selective dehydration of 1,4-butanediol to 3-buten-1-ol and also of 1,3-butanediol to form 3-buten-2-ol and 2-buten-1-ol, while the zirconate catalysts were less active in the dehydration of 2,3-butanediol. The calcination of $Y_2Zr_2O_7$ significantly affected the catalytic activity of the dehydration of 1,4-butanediol: a calcination temperature of $Y_2Zr_2O_7$ at 900 °C or higher was efficient for selective formation of unsaturated alcohols. $Y_2Zr_2O_7$ with high crystallinity exhibits the highest productivity of 3-buten-1-ol from 1,4-butanediol at 325 °C.

**Keywords:** dehydration; butanediol; unsaturated alcohols; $Y_2Zr_2O_7$; cubic fluorite

## 1. Introduction

Biomass has been regarded as an alternative resource to manufacture useful chemicals in the petrochemical industry. Butanediols (BDOs) such as 2,3-butanediol (2,3-BDO), 1,4-butanediol (1,4-BDO) and 1,3-butanediol (1,3-BDO) are derivable in the direct microbial conversion of renewable materials, and they are summarized in several reviews [1–7]. The microbial production of 2,3-BDO is most widely investigated among the BDO isomers and has a long history over a century [1]. 2,3-BDO is produced from corncob and kenaf core as well as sugars such as glucose, xylose, and sucrose. The production of 1,4-BDO from biomass has developed rapidly [8–11]: Yim et al. first reported the direct production of 1,4-BDO from glucose by using *Escherichia coli* strains [8]. 1,4-BDO can be also produced through the chemical process of hydrogenation of biomass-derived succinic acid [12]. 1,3-BDO can be produced from glucose using *Escherichia coli* [13,14]. 1,3-BDO can be also produced from 4-hydroxy-2-butanone, which is derived from biomass-derived levulinic acid, through microbial fermentation [15,16]. The biomass-derived BDOs can be utilized to produce chemicals such as 1,3-butadiene (BD), 3-buten-1-ol (3B1ol), tetrahydrofuran (THF), etc., which are summarized in recent reviews [17–20].

The direct production of BD from BDOs has been reported [21–26]. Phosphate catalysts are effective for the BD production from 2,3-BDO [21,22]. Acidic catalysts such as zeolites and mesoporous

solid acids are active for the BD formation from 1,3-BDO [23–26] while propylene is produced as a major by-product. In our pioneering research, it has been found that 1,4-BDO is preferentially dehydrated to 3B1ol in the vapor phase over $ZrO_2$ catalysts [27,28]. Pure rare earth oxides such as $CeO_2$, $Er_2O_3$, and $Yb_2O_3$ are also active for the reaction and more selective to 3B1ol than $ZrO_2$ catalysts [17,29,30]. Zhang et al. reported an active $CaO$-$ZrO_2$ acid-base catalyst for the dehydration of 1,4-BDO [31]. $ZrO_2$-supported $Yb_2O_3$ and CaO also show high selectivity to 3B1ol [32,33]. We recently investigated the catalytic dehydration of 1,4- and 1,3-BDO over yttria-stabilized tetragonal zirconia (YSZ) catalysts [34]. In the dehydration of 1,4-BDO, a crystalline YSZ with a $Y_2O_3$ content of 3.2 wt.% exhibits an excellent stable catalytic activity: the highest selectivity to 3B1ol of 75.3% at the 1,4-BDO conversion of 26.1% is obtained at 325 °C. The YSZ catalyst also shows an excellent catalytic activity in the dehydration of 1,3-BDO: the highest selectivity to unsaturated alcohols (UOLs) such as 3-buten-2-ol (3B2ol) and 2-buten-1-ol (2B1ol) over 98% is attained at a conversion of 66%.

We have reported an excellent $Yb_2Zr_2O_7$ catalyst with high crystalline of oxygen-defected fluorite for the vapor-phase dehydration of 1,3-BDO to produce UOLs [35]. The sample prepared through a hydrothermal (HT) process in an ammonia media is confirmed to be oxygen-defected type cubic fluorite, $Yb_2Zr_2O_7$, while the as-prepared co-precipitate sample is amorphous. The $Yb_2Zr_2O_7$ maintains a high specific surface area (*SA*) as ca. 40 $m^2\ g^{-1}$ even after being calcined at temperatures higher than 800 °C, in contrast to the fact that catalysts without HT are readily sintered at the temperatures. The $Yb_2Zr_2O_7$ calcined at 900 °C shows the best catalytic performance: 1,3-BDO conversion of 82% is achieved with the total selectivity to UOLs higher than 95% at 325 °C. Also, Fang et al. have recently reported that $Ln_2Zr_2O_7$ (Ln = La, Pr, and Sm) pyrochlores as well as $Y_2Zr_2O_7$ fluorite show low-temperature activity for the oxidative coupling of methane [36]. Thus, we expected that rare earth zirconate (REZrO) samples including $Yb_2Zr_2O_7$ would have a great possibility as a catalyst to activate glycol compounds and to produce UOLs through selective dehydration of BDOs.

In this study, we focused on the efficient production of UOLs from 1,4-BDO as well as BDO isomers such as 1,3- and 2,3-BDO (Figure 1), and we performed catalyst screening in 16 REZrO samples. In the preliminary screening, we have found out $Y_2Zr_2O_7$ as a promising catalyst in the dehydration of 1,4- and 1,3-BDO to produce 3B1ol and a mixture of 2B1ol and 3B2ol, respectively. Detail reaction conditions that $Y_2Zr_2O_7$ works efficiently in the catalytic dehydration of 1,4-BDO were estimated, and reaction mechanisms for the dehydration of 1,4-BDO over $Y_2Zr_2O_7$ were discussed.

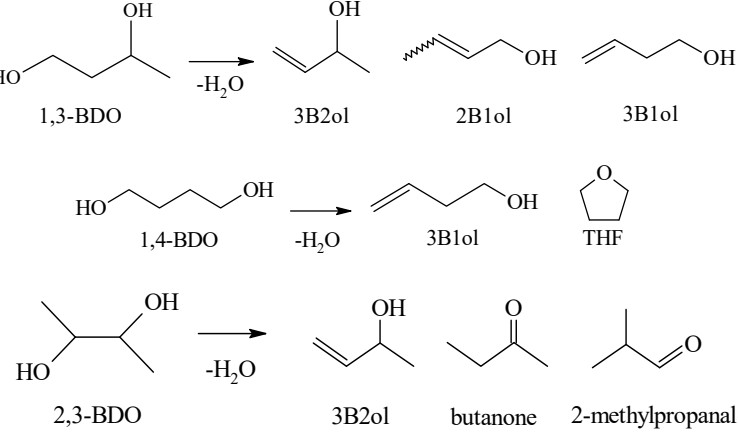

**Figure 1.** Dehydration of butanediols.

## 2. Results

### 2.1. Structural Property of REZrO Catalysts

Various REZrO samples were prepared by HT treatment using ammonia as a precipitant and a mineralizer. Table 1 lists *SA* of REZrO samples calcined at 900 °C. The *SA* values were in the range

between 16 and 39 m$^2$ g$^{-1}$. To confirm that the composition of RE/Zr, XRF analyses of REZrOs were performed, and the results are also summarized in Table 1. A typical N$_2$ adsorption-desorption isotherm of YZrO is shown in Figure S1. In most of the samples except LaZrO, PrZrO, CeZrO, and ScZrO, the ratios of RE/Zr were close to 1 or a little larger, and one of the reasons is that Zr reagent has Hf as an impurity. This indicates that the REZrO samples have a composition of RE/Zr = ca. 1. LaZrO, PrZrO, CeZrO, and ScZrO samples, however, had a RE/Zr ratio less than 1.

**Table 1.** Properties of REZrO catalysts calcined at 900 °C.

| Catalyst | Ionic Radius of Rare Earth Cation [a] (nm) | Average Ionic Radius, R [b] (nm) | $SA$[c] (m$^2$ g$^{-1}$) | Molar Ratio of RE/Zr/Hf [d] | Ratio of RE/Zr [d] |
|---|---|---|---|---|---|
| LaZrO | 0.1032 | 0.0936 | 16 | 43.3/56.1/0.6 | 0.77 |
| PrZrO | 0.0990 | 0.0915 | 29 | 45.1/54.0/0.9 | 0.84 |
| NdZrO | 0.0983 | 0.09115 | 25 | 52.8/46.6/0.6 | 1.13 |
| CeZrO | 0.0970 | 0.0905 | 30 | 47.1/52.3.0.6 | 0.90 |
| SmZrO | 0.0958 | 0.0899 | 32 | 52.9/46.7/0.4 | 1.13 |
| EuZrO | 0.0947 | 0.08935 | 35 | 51.0/48.4/0.6 | 1.05 |
| GdZrO | 0.0938 | 0.0889 | 35 | 52.2/47.1/0.7 | 1.11 |
| TbZrO | 0.0923 | 0.08815 | 34 | 51.8/47.6/0.6 | 1.09 |
| DyZrO | 0.0912 | 0.0876 | 32 | 53.2/46.2/0.6 | 1.15 |
| HoZrO | 0.0901 | 0.08705 | 30 | 53.5/45.9/0.6 | 1.17 |
| YZrO | 0.0900 | 0.0870 | 36 | 51.2/48.0/0.8 | 1.07 |
| ErZrO | 0.0890 | 0.0865 | 30 | 54.1/45.5/0.4 | 1.19 |
| TmZrO | 0.0880 | 0.0860 | 39 | 53.4/46.2/0.4 | 1.16 |
| YbZrO | 0.0868 | 0.0854 | 27 | 55.0/44.6/0.4 | 1.23 |
| LuZrO | 0.0861 | 0.08505 | 26 | 52.8/47.0/0.2 | 1.12 |
| ScZrO | 0.0745 | 0.07925 | 23 | 27.2/71.8/1.0 | 0.38 |

[a] Ionic radius of trivalent rare earth cation with 6 coordination except that of Ce$^{4+}$ with 8 coordination. The data are cited from Ref. [37]. [b] Average ionic radius between Zr$^{4+}$ and rare earth cation. [c] Specific surface area. [d] Estimated by XRF. An average value in five measurements.

Figure 2 depicts continuous scanning XRD patterns of 16 REZrO samples prepared through HT process using ammonia followed by calcination at 900 °C, where RE = La, Ce, Pr, Nd, Sm, Eu, Gd, Tb, Dy, Ho, Y, Er, Tm, Yb, Lu, and Sc. In the zirconates of light rare earth such as La, Ce, Pr, Nd, Sm, and Eu together with GdZrO, the samples were composed of a zirconate and a separated rare earth oxide. For example, the XRD pattern of CeZrO fitted with the patterns of tetragonal Zr$_{0.88}$Ce$_{0.12}$O$_2$ (PDF 01-082-1398, *P4$_2$/nmc*, No. 137) and cubic fluorite-type Ce$_{0.91}$Zr$_{0.0.9}$O$_2$ (PDF 01-075-9496, *Fm−3m*, No. 225) phases, and EuZrO had two phases such as cubic defect fluorite-type Eu$_2$Zr$_2$O$_7$ (PDF 01-078-1292, *Fm−3m*, No. 225) and cubic bixbyite Eu$_2$O$_3$ (PDF 01-073-6281, *I2$_1$3*, No. 199). In addition, no monoclinic ZrO$_2$ phase was observed in the REZrO samples, whereas pure ZrO$_2$ prepared in the HT process without rare earth elements has a monoclinic structure [34]. Although La$_2$O$_3$-ZrO$_2$, Nd$_2$O$_3$-ZrO$_2$, and Sm$_2$O$_3$-ZrO$_2$ prepared in HT process using KOH as a mineralizer are pyrochlore-type RE$_2$Zr$_2$O$_7$ (*Fd−3m*, No. 227), for example, La$_2$Zr$_2$O$_7$ (PDF 01-070-5602) and Sm$_2$Zr$_2$O$_7$ (PDF 00-024-1012) [35], LaZrO, NdZrO, and SmZrO were composed of different phases in the present paper using ammonia as a mineralizer. In contrast to the light rare earth zirconates, REZrOs of heavy rare earth such as TbZrO, DyZrO, HoZrO, YZrO, ErZrO, TmZrO, YbZrO, and LuZrO were composed of a single phase of zirconate, cubic fluorite-type RE$_2$Zr$_2$O$_7$ such as Dy$_2$Zr$_2$O$_7$ (PDF 01-078-1293), Ho$_2$Zr$_2$O$_7$ (PDF 01-078-1294), Er$_2$Zr$_2$O$_7$ (PDF 01-078-1299), and Yb$_2$Zr$_2$O$_7$ (PDF 01-078-1300). However, YZrO was not assigned; both pyrochlore-type (PDF 01-074-9311) and fluorite-type Y$_2$Zr$_2$O$_7$ (PDF 01-081-8080) were possible, as shown in Figure S2. To confirm the structure of YZrO, Rietveld analysis was performed in Section 2.3. Among the REZrOs, the ScZrO sample has a rhombohedral structure, Zr$_3$Sc$_4$O$_{12}$ (PDF 01-071-1022, *R*-3, No. 148). The XRD results indicate that the ScZrO sample has Sc-rich composition.

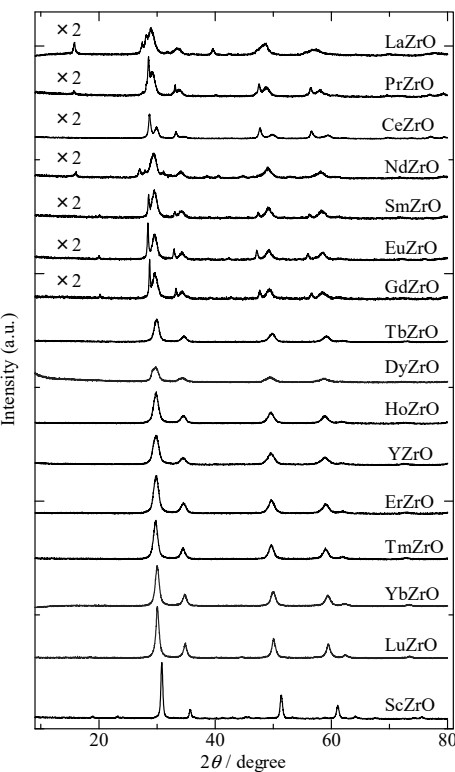

**Figure 2.** XRD patterns of REZrO samples calcined at 900 °C.

## 2.2. Dehydration of Different BDOs over Various REZrO Catalysts

The dehydration of three BDOs such as 1,4-, 1,3-, and 2,3-BDO was performed over REZrO catalysts at a reaction temperature of 325 °C and a space time, *W/F*, of 0.31 h where *W* and *F* are the catalyst weight (g) and the flow rate of BDO (g h$^{-1}$), respectively. Table 2 shows the results in the dehydration of 1,4-BDO over 16 REZrO catalysts calcined at 900 °C. Six REZrOs such as LaZrO, PrZrO, NdZrO, CeZrO, SmZrO, and ScZrO showed low conversions of 1,4-BDO below 40% with the selectivity to 3B1ol lower than 66%. Over LaZrO, NdZrO, and ScZrO, THF was the major product. In contrast, the other ten REZrO catalysts showed the 1,4-BDO conversions higher than 50% with the 3B1ol selectivity over 75%. Among the REZrOs, the selectivity to the total of 3B1ol, 2B1ol, and BD, which maintained a straight C4 carbon chain structure, exceeded 90% over DyZrO, YZrO, and ErZrO. In particular, YZrO showed the highest conversion of 77.6% as well as the highest 3B1ol selectivity of 87.7%. It was confirmed that the catalytic activity of YZrO was higher than the most active CaO/ZrO$_2$ catalyst that had been ever reported [33].

Table 3 shows the comparison of reactivity of different BDOs in the dehydration of 1,4-, 1,3-, and 2,3-BDO over several REZrO catalysts calcined at 900 °C under the same reaction conditions as those in Table 2. In the dehydration of 1,3-BDO over 6 selected REZrO catalysts such as LaZrO, CeZrO, YZrO, YbZrO, LuZrO, and ScZrO, YZrO and YbZrO catalysts showed the 1,3-BDO conversions higher than 89% whereas LaZrO and ScZrO showed low conversions below 50%. The catalysts had high selectivity to UOLs over 80%. The sum of selectivity to UOLs and BD exceeded 94%, except LaZrO. In particular, over YZrO, YbZrO, and LuZrO, the sum of selectivity to UOLs and BD exceeded 97%. In the same way as 1,4-BDO, YZrO showed the highest 1,3-BDO conversion of 93.9% with high UOLs selectivity of 90.2%, while the selectivity to BD was the highest.

In the dehydration of 2,3-BDO over several REZrO catalysts calcined at 900 °C (Table 3), under the same reaction conditions, the REZrO catalysts showed 2,3-BDO conversions lower than 20% with the selectivity to 3B2ol at most 50%. Unfortunately, the catalytic activity of REZrO as well as YZrO is inferior to the catalysts such as ZrO$_2$ and Sc$_2$O$_3$ previously reported in the references [38,39]. Thus, in the catalyst screening, it is concluded that the REZrO catalysts, especially YZrO, are active for the

vapor-phase dehydration of 1,4- and 1,3-BDO, but they are not suitable for the activation of 2,3-BDO. In each REZrO catalyst, the highest UOLs yield was obtained in the dehydration of 1,3-BDO and the lowest one in 2,3-BDO. Therefore, the reactivity order of BDOs over REZrO catalysts was 1,3-BDO > 1,4-BDO > 2,3-BDO. Because YZrO showed the highest 1,4-BDO conversion and the highest 3B1ol selectivity, we investigated the YZrO catalyst in detail in the following sections.

**Table 2.** Dehydration of 1,4-BDO over REZrO catalysts calcined at 900 °C.

| Catalyst | Conversion | Selectivity (mol%) | | | | | | |
|---|---|---|---|---|---|---|---|---|
| | (%) | 3B1ol | 2B1ol | UOLs | BD | THF | GBL | Others |
| LaZrO | 29.1 | 6.1 | 0.5 | 6.6 | 0.0 | 88.4 | 3.1 | 1.9 |
| PrZrO | 25.2 | 61.7 | 1.5 | 63.2 | 0.0 | 32.7 | 2.5 | 1.6 |
| NdZrO | 37.7 | 32.4 | 0.8 | 33.2 | 0.0 | 63.7 | 0.5 | 2.6 |
| CeZrO | 24.7 | 60.8 | 5.6 | 66.4 | 0.0 | 24.9 | 5.6 | 3.1 |
| SmZrO | 37.3 | 65.8 | 1.3 | 67.1 | 0.1 | 30.8 | 0.9 | 1.1 |
| EuZrO | 52.8 | 82.6 | 2.8 | 85.4 | 0.8 | 10.8 | 2.2 | 0.8 |
| GdZrO | 68.5 | 81.5 | 1.9 | 83.4 | 0.5 | 15.3 | 0.4 | 0.4 |
| TbZrO | 52.4 | 76.1 | 1.1 | 77.2 | 0.6 | 21.0 | 0.9 | 0.3 |
| DyZrO | 69.4 | 86.5 | 2.7 | 89.2 | 1.6 | 7.9 | 0.4 | 0.9 |
| HoZrO | 51.2 | 82.6 | 1.7 | 84.3 | 0.8 | 13.3 | 0.7 | 0.9 |
| YZrO | 77.6 | 87.7 | 2.8 | 90.5 | 1.2 | 7.5 | 0.6 | 0.2 |
| ErZrO | 64.4 | 85.8 | 4.0 | 89.8 | 1.9 | 7.4 | 0.3 | 0.6 |
| TmZrO | 55.9 | 82.8 | 1.3 | 84.1 | 1.1 | 13.3 | 0.3 | 1.2 |
| YbZrO | 61.3 | 84.8 | 2.2 | 87.0 | 0.8 | 11.5 | 0.3 | 0.4 |
| LuZrO | 51.2 | 84.3 | 2.0 | 86.3 | 2.6 | 9.0 | 0.3 | 1.8 |
| ScZrO | 35.5 | 30.0 | 0.1 | 30.1 | 0.0 | 68.9 | 0.3 | 0.7 |

Conversion and selectivity are averaged at time on stream (TOS) between 1–5 h. Reaction conditions: temperature, 325 °C; $W/F$, 0.31 h; catalyst weight, 0.50 g; $N_2$ carrier gas flow rate, 30 cm$^3$ min$^{-1}$. 3B1ol, 3-buten-1-ol; 2B1ol, 2-buten-1-ol; UOLs = 3B1ol + 2B1ol; BD, 1,3-butadiene; THF, tetrahydrofuran; GBL, γ-butyrolactone. Others include ethanol, 1-butanol, and some unidentified products.

**Table 3.** Comparison in the dehydration of 1,3-, 1,4- and 2,3-BDO over REZrO calcined at 900 °C.

| Catalyst | Reactant | Conv. | Selectivity (mol%) | | | | | | | | |
|---|---|---|---|---|---|---|---|---|---|---|---|
| | | (%) | 3B2ol | 3B1ol | 2B1ol | UOLs | BD | MEK | THF | 3H2BO | Others |
| LaZrO | 1,3-BDO | 40.2 | 44.9 | 1.2 | 35.1 | 81.2 | 1.7 | 6.8 | - | - | 10.3 [a] |
| | 1,4-BDO | 29.1 | - | 6.1 | 0.5 | 6.6 | 0 | - | 88.4 | - | 5.0 [b] |
| | 2,3-BDO | 4.8 | 28.3 | - | - | 28.3 | 0 | 16.1 | - | 37.1 | 18.5 [c] |
| CeZrO | 1,3-BDO | 64.7 | 55.8 | 1.0 | 35.1 | 91.9 | 2.8 | 0.7 | - | - | 4.6 [a] |
| | 1,4-BDO | 24.7 | - | 60.8 | 5.6 | 66.4 | 0 | - | 24.9 | - | 8.7 [b] |
| | 2,3-BDO | 6.6 | 15.4 | - | - | 15.4 | 0 | 29.4 | - | 33.4 | 21.8 [c] |
| YZrO | 1,3-BDO | 93.9 | 52.1 | 1.3 | 36.8 | 90.2 | 7.2 | 0.2 | - | - | 2.4 [a] |
| | 1,4-BDO | 77.6 | - | 87.7 | 2.8 | 90.5 | 1.2 | - | 7.5 | - | 0.8 [b] |
| | 2,3-BDO | 17.0 | 30.3 | - | - | 30.3 | 0 | 10.9 | - | 26.8 | 32.0 [c] |
| YbZrO | 1,3-BDO | 89.8 | 53.9 | 1.4 | 40.9 | 96.2 | 1.0 | 1.0 | - | - | 1.8 [a] |
| | 1,4-BDO | 61.3 | - | 84.8 | 2.2 | 87.0 | 0.8 | - | 11.5 | - | 0.7 [b] |
| | 2,3-BDO | 12.2 | 46.9 | - | - | 46.9 | 0 | 7.3 | - | 28.4 | 17.4 [c] |
| LuZrO | 1,3-BDO | 83.0 | 50.8 | 1.7 | 42.9 | 95.4 | 2.0 | 0.9 | - | - | 1.7 [a] |
| | 1,4-BDO | 51.2 | - | 84.3 | 2.0 | 86.3 | 2.6 | - | 9.0 | - | 2.1 [b] |
| | 2,3-BDO | 11.3 | 50.8 | - | - | 50.8 | 0 | 10.1 | - | 24.2 | 14.9 [c] |
| ScZrO | 1,3-BDO | 20.6 | 43.1 | 12.7 | 39.7 | 95.5 | 0 | 1.4 | - | - | 3.1 [a] |
| | 1,4-BDO | 35.5 | - | 30.0 | 0.1 | 30.1 | 0 | - | 68.9 | - | 1.0 [b] |
| | 2,3-BDO | 18.4 | 47.4 | - | - | 47.4 | 0 | 18.7 | - | 16.0 | 17.9 [c] |

Reaction conditions are the same as those in Table 2. 3B2ol, 3-buten-2-ol; 3B1ol, 3-buten-1-ol; 2B1ol, 2-buten-1-ol; UOLs = 3B2ol+3B1ol+2B1ol; BD, 1,3-butadiene; MEK, butanone; THF, tetrahydrofuran; 3H2BO, 3-hydroxy-2-butanone. [a] Others include ethanol, acetone, 1-butanol, 3-buten-2-one, etc. [b] Others include γ-butyrolactone, 1-butanol, ethanol, etc. [c] Others include 2-methylpropanal, 2-methyl-1-propanol, 2-butanol, ethanol, acetone, etc.

### 2.3. Effect of Calcination Temperature on the Structure of YZrO Catalysts and Catalytic Activity in the Dehydration of 1,4-BDO

Table 4 lists *SA* of YZrO samples calcined at different temperatures. The *SA* value was decreased when the raising calcination temperature from 91 to 16 m$^2$ g$^{-1}$ at the temperature range between 600 and 1050 °C. The particle size calculated from the *SA*, $D_{BET}$, is also summarized in Table 4. Figure 3 shows continuous scanning XRD analysis of YZrO samples calcined at different temperatures. The as-prepared sample was crystallized during the HT process. All the samples calcined at 600–1050 °C had a major diffraction peak at 29.9 degree. The crystallite size, $D_{XRD}$, which was estimated from of the major diffraction peak using the Scherrer equation, is also summarized. The $D_{XRD}$ value was simply increased when raising the calcination temperature. There is a significant difference between the crystallite size and the particle size: $D_{XRD}$ was much smaller than $D_{BET}$. To confirm the morphology of catalysts calcined at different temperatures.

**Table 4.** Physical properties of YZrO calcined at different temperatures.

| Calcination (°C) | SA (m$^2$ g$^{-1}$) | FWHM at 2$\theta$ = 29.9° (degree) | Particle Size, $D_{BET}$ (nm) | Crystallite Size, $D_{XRD}$ (nm) |
|---|---|---|---|---|
| 600 | 91 | 1.792 | 12 | 4.6 |
| 800 | 51 | 1.439 | 22 | 5.7 |
| 900 | 36 | 1.242 | 31 | 6.6 |
| 1000 | 19 | 0.670 | 58 | 12 |
| 1050 | 16 | 0.543 | 70 | 15 |

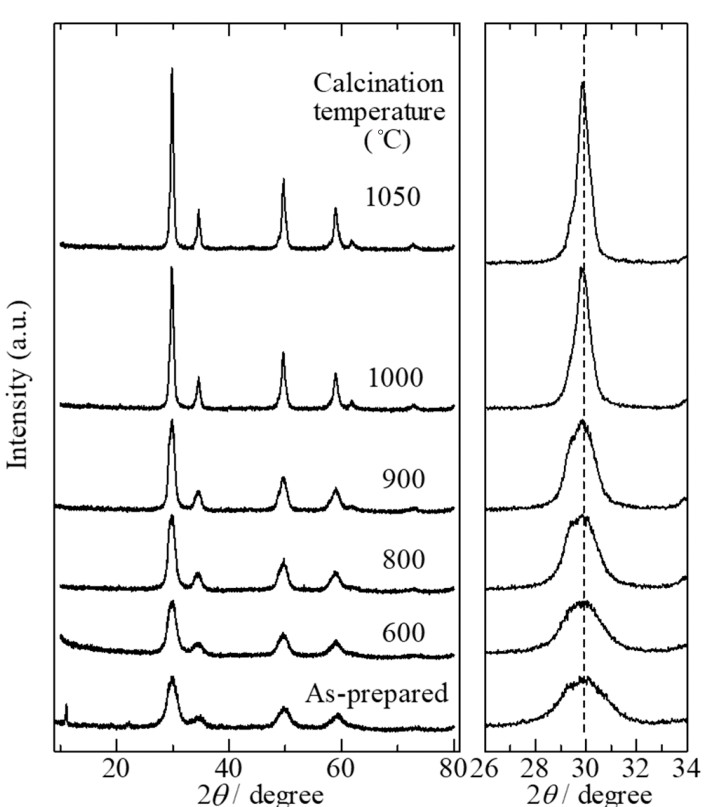

**Figure 3.** Continuous scanning XRD patterns of YZrO.

For YZrO, we observed the FE-SEM image (Figure 4). The SEM image clearly shows that the YZrO sample calcined at 900 °C is composed of agglomerates of particles with the size of 30–50 nm, which is consistent with the $D_{BET}$ value of 31 nm rather than the $D_{XRD}$ of 6.6 nm. This indicates that the particles shown in the SEM image are non-porous primary particles.

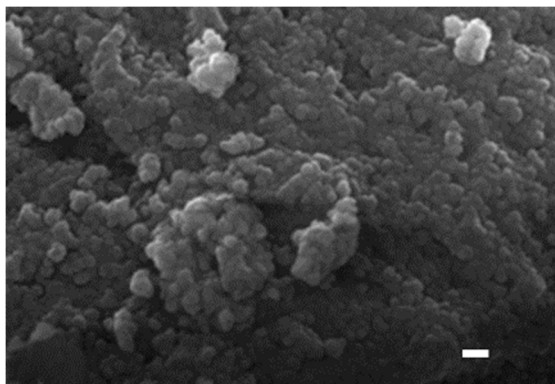

**Figure 4.** FE-SEM image of YZrO calcined at 900 °C. A scale bar in the photo indicates 100 nm.

Even at a high calcination temperature of 1050 °C, the result of Rietveld analysis was not satisfied within the fitting errors because of low diffraction intensity. Figure 5 shows the step-scanning XRD pattern of YZrO sample calcined at 1200 °C. We confirmed diffraction angles are the same between YZrO calcined at 1050 and 1200 °C. Thus, Rietveld analysis was performed in a YZrO sample calcined at 1200 °C to refine the structure parameter (Figure 5). We confirmed that the sample was composed of defect fluorite-type $Y_2Zr_2O_7$, but not pyrochlore. Table 5 lists the refined structure parameters of $Y_2Zr_2O_7$.

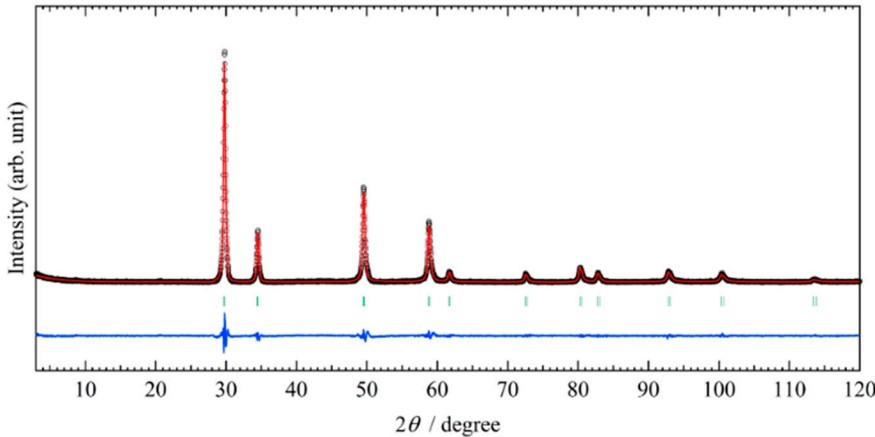

**Figure 5.** Step-scanning XRD pattern of YZrO calcined at 1200 °C with calculated XRD pattern.

**Table 5.** Refined structure parameters of $Y_2Zr_2O_7$ and YZrO calcined at 1200 °C.

| Sample | | $Y_2Zr_2O_7$ | |
|---|---|---|---|
| **Space Group** | | *Fm−3m* [a] | |
| $a$ (Å) | 5.19878 | $B$ (Y) | 1.81615 |
| $\alpha$ (degree) | 90.0000 | $B$ (Zr) | 1.77582 |
| $\beta$ (degree) | 90.0000 | $B$ (O) | 3.46843 |
| $\gamma$ (degree) | 90.0000 | $R_{wp}$ (%) | 12.229% |
| $g$ (Y) [b] | 0.5000 | $R_B$ (%) | 2.039% |
| $g$ (Zr) [b] | 0.5000 | $R_F$ (%) | 1.593% |
| $g$ (O) [b] | 0.8750 | $S$ | 1.3891 |

$g$: occupancy; $B$: isotropic atomic displacement parameter. [a] Atomic sites: (Y, Zr) 4*a* (0, 0, 0) and O 8*c* (1/4, 1/4, 1/4).
[b] Fixed during the refinement.

Black points and red lines represent observed and calculated patterns, respectively. Green vertical marks indicate the Bragg reflection positions of defected-fluorite $Y_2Zr_2O_7$ phase. Blue line at the

bottom shows the difference between the observed and calculated patterns (For interpretation of the references to color in this figure legend, the reader is referred to the web version of this article).

Figure 6 shows the effect of calcination temperature of YZrO on the conversion of 1,4-BDO, and the numerical data are located in Table S1. The 1,4-BDO conversion was maximized at 900 °C, and the selectivity to 3B1ol increased with raising the calcination temperature. The selectivity to THF showed an opposite trend to the selectivity to 3B1ol. The conversion, which increases with raising calcination temperature, is in reverse proportion to specific surface area at the range between 600 and 900 °C.

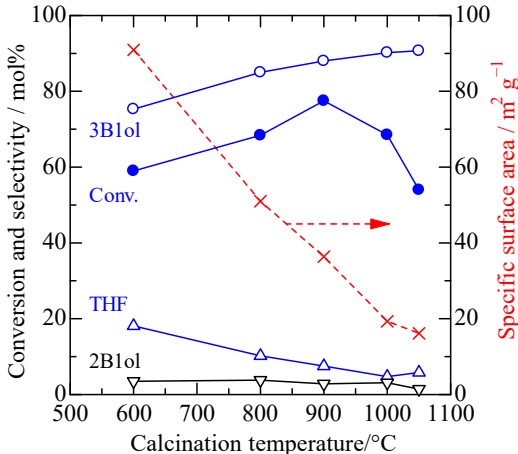

**Figure 6.** Effect of calcination temperature of YZrO catalyst on the dehydration of 1,4-BDO and *SA*.

## 2.4. Catalytic Dehydration of 1,4-BDO over YZrO Calcined at 900 °C under Different Reaction Conditions

Figure 7 shows the change in conversion with different space time at 325 °C. The space time, *W/F*, was controlled by the variation of *W* at a constant *F*, where the numerical data are listed in Table S2. The 1,4-BDO conversion simply increased with increasing the space time. At a *W/F* = 0.94 h, 99.4% of the 1,4-BDO conversion was attained. The selectivity to 3B1ol gradually decreased from 87.7 to 77.9% with the increase in the selectivity to 2B1ol and BD, while the sum of selectivity to UOLs and BD seemed to be constant. This means that both 2B1ol and BD are regarded as the secondary products produced from 3B1ol. The selectivity to THF, however, was almost constant in the variation of *W/F*. Therefore, 3B1ol and THF are the primary products in the dehydration of 1,4-BDO. In addition, the UOLs formation rate at 325 °C was almost constant at conversions lower than 80% (Table S2): the rate is reduced at a *W/F* = 0.31 h by 10% by comparing at 0.19 h. At high conversions near 100%, 2B1ol and BD increased with decreasing 3B1ol (Figure 7 and Table 6).

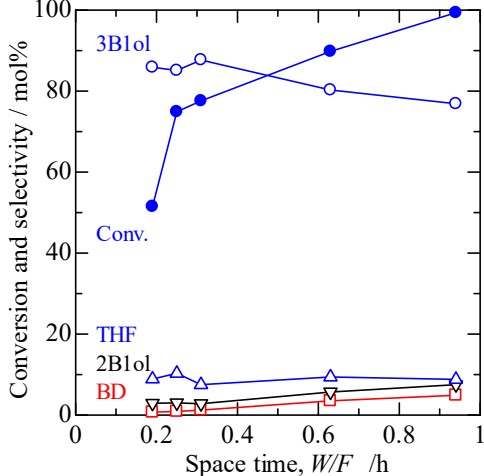

**Figure 7.** Change in the conversion of 1,4-BDO with space time over YZrO calcined at 900 °C.

**Table 6.** Dehydration of 1,4-BDO over YZrO at different reaction temperatures.

| Temperature | Conv. | Selectivity (mol%) | | | | | | | |
|---|---|---|---|---|---|---|---|---|---|
| (°C) | (%) | 3B1ol | 2B1ol | UOLs | BD | Propylene | THF | GBL | Others |
| 300 | 25.9 | 84.0 | 1.1 | 85.1 | 0.4 | 0.0 | 12.4 | 0.6 | 1.5 |
| 325 | 77.6 | 87.7 | 2.8 | 90.5 | 1.2 | 0.0 | 7.5 | 0.6 | 0.2 |
| 350 | 99.8 | 70.2 | 7.1 | 77.3 | 11.5 | 0.0 | 9.9 | 0.2 | 1.1 |
| 360 | 100.0 | 53.0 | 10.1 | 63.1 | 22.2 | 0.0 | 11.5 | 0.1 | 3.1 |
| 375 | 100.0 | 23.8 | 4.4 | 28.2 | 58.8 | 5.8 | 3.7 | 0.1 | 3.4 [a] |

The catalyst sample was calcined at 900 °C. Conversion and selectivity were averaged at TOS between 1–5 h. Reaction conditions: $W/F$, 0.31 h; catalyst weight, 0.50 g; $N_2$ flow rate, 30 cm$^3$ min$^{-1}$. 3B1ol, 3-buten-1-ol; 2B1ol, 2-buten-1-ol; UOLs = 3B1ol + 2B1ol; BD, 1,3-butadiene; THF, tetrahydrofuran; GBL, $\gamma$-butyrolactone. Others include ethanol, 1-butanol, $CO_2$, and some unidentified products. [a] includes 1.8 mol% of $CO_2$.

Table 6 summarized the conversion data at different reaction temperatures in the dehydration of 1,4-BDO over YZrO calcined at 900 °C. With raising the reaction temperature, the conversion was increased, and the complete conversion was attained at 360 °C. The selectivity to 3B1ol was decreased through the maximum at 325 °C: the selectivity to 3B1ol and UOLs was 87.7 and 90.5%, respectively. At a temperature of 350 °C or higher, BD was produced together with the production of propylene, which was formed via the decomposition of 3B1ol [40,41]. The propylene produced at 375 °C was comparable to that of $CO_2$: the selectivity to $CO_2$ at 375 °C was 1.8%, which indicates that a molar ratio of propylene/$CO_2$ is 1.07.

## 2.5. Effect of Carrier Gas on the Catalytic Dehydration of 1,4-BDO over YZrO Calcined at 900 °C

The effect of carrier gas in the dehydration of 1,4-BDO over YZrO catalyst was investigated to clarify the role of basic and acidic sites of YZrO catalyst: either $CO_2$, $NH_3$, or $H_2$ was adapted as a carrier gas instead of $N_2$ gas. Table 7 shows the effect of carrier gas on the conversion of 1,4-BDO over YZrO at 325 and 300 °C. In $CO_2$ flow, the conversion, the 3B1ol selectivity, and the 3B1ol productivity were smaller than those in $N_2$ flow. Thus, $CO_2$ gas could work as a poison of basic sites. In $NH_3$ flow at 325 °C, in contrast, the conversion level and the 3B1ol productivity were slightly higher than those in $N_2$ flow: an $NH_3$ seems to act as an accelerator, but not as a poison. It is much clearer, at 300 °C, that $NH_3$ gas accelerates the reaction. An $NH_3$ gas may act as a base catalyst together with catalyst surface for the abstraction of hydrogen from 1,4-BDO. It is possible that either weakly adsorbed $NH_3$ or gas-phase one may act as a catalyst.

**Table 7.** Effects of carrier gas on the dehydration of 1,4-BDO over YZrO calcined at 900 °C.

| Carrier Gas | Conversion | Selectivity (mol%) | | | | | | | Productivity of 3B1ol |
|---|---|---|---|---|---|---|---|---|---|
| | (%) | 3B1ol | 2B1ol | UOLs | BD | THF | GBL | Others | mol h$^{-1}$ kg$^{-1}$ |
| *T* = 325 °C | | | | | | | | | |
| $N_2$ | 77.6 | 87.7 | 2.8 | 90.5 | 1.2 | 7.5 | 0.6 | 0.2 | 24.2 |
| $CO_2$ | 74.6 | 64.6 | 1.5 | 66.1 | 0.3 | 32.3 | 0.8 | 0.5 | 17.1 |
| $NH_3$ | 81.9 | 85.5 | 6.5 | 92.0 | 0.0 | 6.8 | 0.1 | 1.1 | 24.9 |
| $H_2$ | 87.8 | 86.7 | 3.3 | 90.0 | 1.0 | 8.4 | 0.4 | 0.2 | 27.0 |
| *T* = 300 °C | | | | | | | | | |
| $N_2$ | 25.9 | 84.0 | 1.1 | 85.1 | 0.4 | 12.4 | 0.6 | 1.5 | 7.7 |
| $CO_2$ | 19.2 | 60.8 | 1.5 | 62.3 | 0.0 | 34.0 | 1.6 | 2.1 | 4.1 |
| $NH_3$ | 32.3 | 87.6 | 1.5 | 89.1 | 0.0 | 9.9 | 0.2 | 0.7 | 10.1 |
| $N_2$ ($NH_3$) [a] | 16.0 | 76.6 | 10.3 | 86.9 | 0.0 | 7.8 | 1.5 | 3.8 | 4.4 |

Conversion and selectivity are averaged at TOS between 1–5 h. Reaction conditions: *T*, reaction temperature; $W/F$, 0.31 h; carrier gas flow rate, 30 cm$^3$ min$^{-1}$. 3B1ol, 3-buten-1-ol; 2B1ol, 2-buten-1-ol; UOLs = 3B1ol + 2B1ol; BD, 1,3-butadiene; THF, tetrahydrofuran; GBL, $\gamma$-butyrolactone. Others include ethanol, 1-butanol, and some unidentified products. [a] Prior to the preheating in $N_2$ at 300 °C for 1 h, $NH_3$ was flowed at a rate of 30 cm$^3$ min$^{-1}$ and 300 °C for 0.5 h.

Then, we adopted pre-adsorption of NH$_3$ before the reaction to confirm the above-mentioned hypothesis. In Figure S3, the conversion at TOS of the initial 2 h was higher than 25%, which is higher than that in N$_2$, but it reduced to 13% after TOS of 2 h. This indicates either hydrogen-bonded or weakly adsorbed molecular NH$_3$ acts as a base catalyst, but it removes from the surface during the reaction. Thus, NH$_3$ adsorbed on YZrO at 300 °C acted as a poison of acidic sites in N$_2$ flow (Table 7). In the formation of THF, however, the selectivity to THF was preferential in CO$_2$ flow, and it was depressed in NH$_3$ flow at both 325 and 300 °C. In H$_2$ flow, the 1,4-BDO conversion was higher than that in N$_2$ flow at 325 °C. H$_2$ gas accelerates the formation of 3B1ol in the dehydration of 1,4-BDO over YZrO catalyst. The acceleration effect of H$_2$ will be discussed in Section 3.3.

Figure 8 shows the changes in the conversion of 1,4-BDO with time on stream in long run tests. The catalytic activity, i.e., the 1,4-BDO conversion and the selectivity to each product, was stabilized at a conversion of ca. 87% in H$_2$ flow for 30 h. In N$_2$ flow, however, a gradual decrease in the 1,4-BDO conversion was observed; the conversion was decreased from 79 to 75% for 30 h. As shown in Table 7, the 1,4-BDO conversion in N$_2$ was lower than that in H$_2$ although the selectivity to 3B1ol was higher. The highest 3B1ol productivity of 27.0 mol h$^{-1}$ kg$^{-1}$ was obtained in H$_2$ flow, while it was 24.2 mol h$^{-1}$ kg$^{-1}$ in N$_2$ flow.

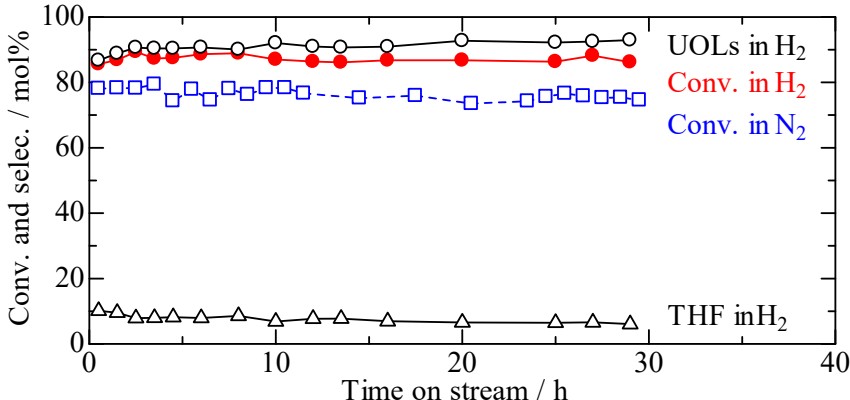

**Figure 8.** Long-run stability in the conversion of 1,4-BDO over YZrO calcined at 900 °C. Reaction temperature, 325 °C; carrier gas flow rate, 30 cm$^3$ min$^{-1}$; *W/F* = 0.31 h.

*2.6. Acid-base Properties of YZrO Catalyst Calcined at 600 and 900 °C*

Figure 9 illustrates temperature-programmed desorption (TPD) profiles of adsorbed NH$_3$ and CO$_2$ on YZrO calcined at 600 and 900 °C. In Figure 9A, the amount of desorbed CO$_2$ from YZrO calcined at 600 °C is more than that of NH$_3$. The amounts of acidic and basic sites were calculated to be 0.193 and 0.313 mmol g$^{-1}$ up to 600 °C, respectively. In contrast, the TPD profiles of NH$_3$ and CO$_2$ adsorbed on YZrO calcined at 900 °C were quite similar to each other (Figure 9B): the amounts of acidic and basic sites were calculated to be 0.123 and 0.127 mmol g$^{-1}$ up to 500 °C, respectively.

Acid strength distributions obtained for both catalysts showed little variations in the low temperature range. In the high temperature range above 400 °C, the reduction of NH$_3$ released during TPD for the sample calcined at 900 °C suggests a slight reduction of the stronger acid sites when rising the calcination temperature from 600 to 900 ºC. In addition, the total number of acidic sites was clearly reduced, when rising the calcination temperature. Base strength distribution showed the same trend as the acid strength distribution, when comparing both YZrO catalysts. Acidic and basic sites on which NH$_3$ and CO$_2$ adsorbed at temperatures higher than 325 °C are considered to be poisoned in the reaction at 325 °C. As long as strong acidic and strong basic sites measured by the desorption at >325 °C, the amounts of acidic and basic sites of YZrO calcined at 600 °C were calculated to be 0.041 and 0.074 mmol g$^{-1}$, while those of YZrO calcined at 900 °C were 0.015 and 0.022 mmol g$^{-1}$, respectively.

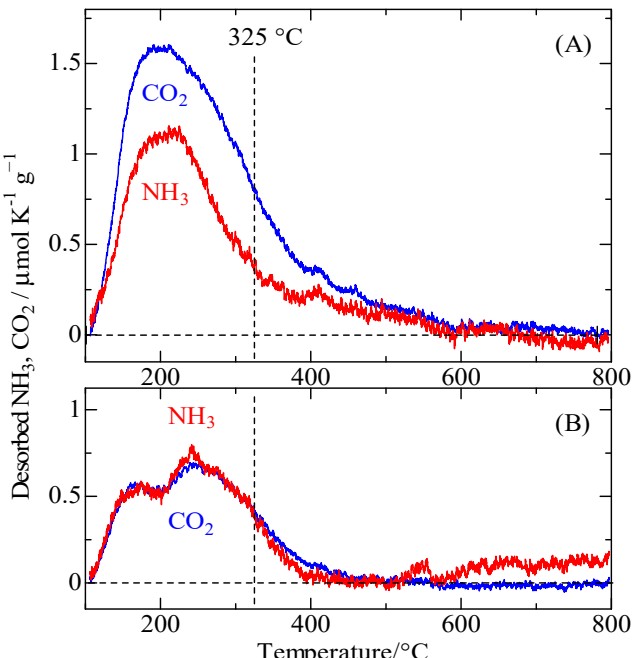

**Figure 9.** TPD profiles of adsorbed $NH_3$ and $CO_2$ on YZrO calcined at 600 (**A**) and 900 °C (**B**).

## 3. Discussion

### 3.1. Structure of REZrO as an Effective Catalyst for the Dehydration of BDOs

In the preparation of zirconate, KOH is frequently used as a mineralizer: $La_2Zr_2O_7$ is synthesized through the HT process of double hydrous oxides of La and Zr using KOH as the mineralizer [42]. Liu et al. have synthesized REZrO ceramics such as $Yb_2Zr_2O_7$ and $Gd_2Zr_2O_7$ at calcination temperatures above 1550 °C [43]. An example of REZrO for catalyst use is rare; Fang et al. recently reported that oxygen-defected fluorite $Y_2Zr_2O_7$ is prepared with a simple co-precipitation method using ammonia as a precipitant: the sample calcined at 800 °C has *SA* as small as 7 m$^2$ g$^{-1}$ and is used for methane oxidative coupling [36]. We previously prepared REZrO samples in HT conditions using KOH solution, but they had small *SA* values at most 10 m$^2$ g$^{-1}$ and showed low conversion in the dehydration of 1,3-BDO [35]. However, $Yb_2Zr_2O_7$ prepared by HT aging using ammonia has a high *SA* of 49 m$^2$ g$^{-1}$ and shows an excellent catalytic activity in the dehydration of 1,3-BDO. Thus, HT aging with ammonia mineralizer would be promising for the preparation of catalyst samples with high crystallinity and high *SA*. In this paper, we obtained REZrO samples with high *SA* between 16 and 39 m$^2$ g$^{-1}$ (Table 1) in HT treatment using ammonia as a mineralizer.

In our previous paper [35], $La_2O_3$-$ZrO_2$, $Nd_2O_3$-$ZrO_2$, and $Sm_2O_3$-$ZrO_2$ prepared in the HT process using KOH as a mineralizer were determined to be pyrochlore-type $Ln_2Zr_2O_7$. The reason is that the ionic radius ratio of Ln/Zr determines the structure of $Ln_2Zr_2O_7$ [44]: it is reported that the pyrochlore-type structure is favored at the ionic radius ratio of Ln/Zr between 1.46 and 1.78, while the defect fluorite-type structure is favorable at a ratio less than 1.46. At the ratio of 1.46, $Gd_2Zr_2O_7$ could have both structures. In the continuous scanning XRD patterns (Figure 2), structures of the samples are assigned by comparing the PDF database. Most of the structural data of REZrOs of heavy rare earth were fitted with the database of cubic fluorite-type $RE_2Zr_2O_7$, as described in Section 2.1. In the case of YZrO, however, the structure was fitted with both pyrochlore and fluorite phases. In Figure 5, Rietveld analysis of the YZrO calcined at 1200 °C clearly indicates the structure is fluorite. It is confirmed that the active YZrO is composed of defect fluorite-type $Y_2Zr_2O_7$, but not pyrochlore. The structure is the same as that of oxygen-defected fluorite $Yb_2Zr_2O_7$, which is reported previously [35].

### 3.2. Dehydration of Different BDOs over Various REZrO Catalysts

In the catalyst screening, it is concluded that the reactivity order of BDOs over each REZrO catalyst is 1,3-BDO > 1,4-BDO > 2,3-BDO. In this work, YZrO is the most active catalyst as a weight basis for the vapor-phase dehydration of 1,4- and 1,3-BDO. We have previously reported efficient rare earth oxide catalysts for the dehydration of 1,4-BDO to produce 3B1ol [17]: rare earth oxides such as $Er_2O_3$, $Yb_2O_3$, and $Lu_2O_3$ show the highest 3B1ol formation rates in the catalytic reaction. The formation rate varies with the ionic radius of the rare earth cation.

Here, we summarized the relationship between the formation rate and the ionic radius of the rare earth cation. Figure 10 shows the intrinsic activity of REZrO catalyst: the formation rate of UOLs per unit surface area with the variation of ionic radius of rare earth cation (Figure 10A) and average ionic radius between $Zr^{4+}$ and rare earth cation, $R_{(RE+Zr)/2}$ (Figure 10B). The formation rate of UOLs per unit surface area was calculated from the data in Table 2, Table 3, and Table S3 for the dehydration of 1,4-, 2,3-, and 1,3-BDO, respectively. In the present study, REZrO catalysts with heavy rare earth cations such as DyZrO, HoZrO, YZrO, ErZrO, YbZrO, and LuZrO show the comparable formation rate of UOLs, YbZrO shows the highest UOLs formation rates among the REZrOs in the dehydration of both 1,3- and 1,4-BDO (Figure 10A). In comparison with pure rare earth oxides, we adapted a mathematical mean of ionic radius between RE and Zr, which is listed in Table 1. The mathematical mean of ionic radius between Gd and Zr, $R_{(Gd+Zr)/2}$, is calculated to be 0.0889 nm and that between Dy and Zr, $R_{(Dy+Zr)/2}$ = 0.0876 nm, where $R_{Gd}$ = 0.0938, $R_{Dy}$ = 0.0912, and $R_{Zr}$ = 0.0840 nm [37]. In Figure 10B, the formation rates over REZrO are larger than those over pure rare earth oxides reported in Ref. [45]. Pure $CeO_2$ has exceptionally high activity in the dehydration of 1,3-BDO; it shows the highest formation rate of 3.85 mmol $h^{-1}$ $m^{-2}$ at the ionic radius of $Ce^{4+}$, 0.0970 nm, as shown in Figure 10B with the dotted line. Active REZrO catalysts appear in a narrow range of ionic radius. The mean ionic radii of 0.0889 nm for GdZrO and 0.0854 nm for YbZrO are close to the ionic radii of $Er^{3+}$ in $Er_2O_3$ and $Lu^{3+}$ in $Lu_2O_3$, which are 0.0890 and 0.0861 nm, respectively [45]. The mean ionic radius of 0.0870 nm, $R_{(Y+Zr)/2}$, for YZrO is close to the ionic radius of $Yb^{3+}$ of 0.0868 in $Yb_2O_3$, which is reported to be an active catalyst for the dehydration of 1,4-BDO [46]. This suggests that crystallites with specific size of unit cell are efficient for the selective formation of 3B1ol. Besides, because yttrium is abundant and cheap among rare earth elements, we investigated the YZrO catalyst in detail.

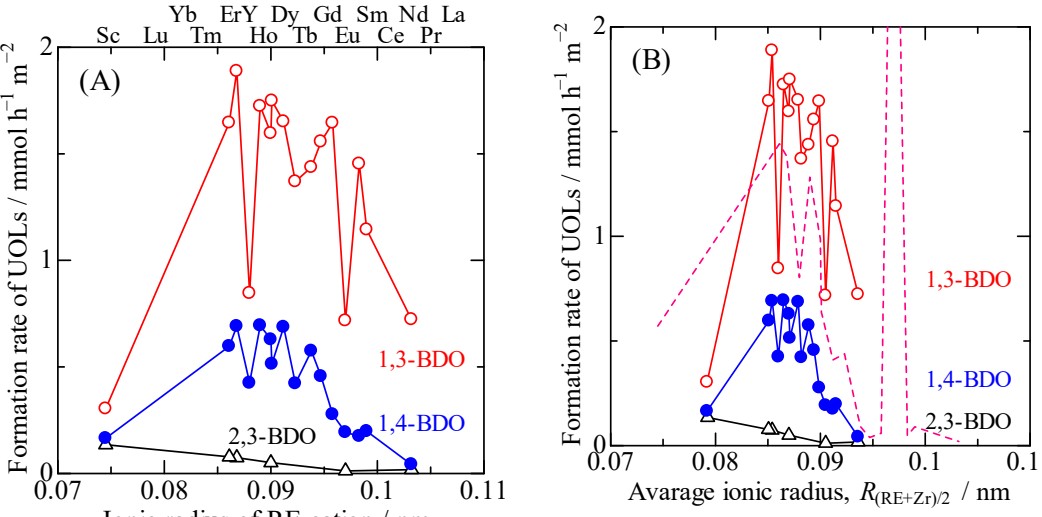

**Figure 10.** Changes in the formation rate of unsaturated alcohols (UOLs) in the dehydration of BDOs over REZrO catalysts at 325 °C with ionic radius of rare earth cation (**A**) and average ionic radius between $Zr^{4+}$ and rare earth cation, $R_{(RE+Zr)/2}$ (**B**). A dotted line in Figure 10B indicates the formation rate in the dehydration of 1,3-BDO over pure rare earth oxide with ionic radius of rare earth cation, reported in Ref. [45].

### 3.3. Significant Parameters on the Catalytic Activity of YZrO

As shown in Figure 6, the calcination temperature of YZrO significantly affects the conversion of 1,4-BDO. The effect of calcination temperature on the 1,4-BDO conversion over YZrO is quite similar to those observed in the reaction over pure rare earth oxides such as $Er_2O_3$, $Lu_2O_3$ [46], and $CeO_2$ [47]. The significant increase in the activity is probably attributed to the change in the ratio of exposed crystal surface with the particle sizes.

In Table 7, $H_2$ carrier gas accelerates the formation of 3B1ol at 325 °C. It has been reported in the literature that the acceleration by $H_2$ in the dehydration of BDOs: 1,4-BDO over $CaO/ZrO_2$ [33] and 2,3-BDO over $Sc_2O_3$ [39]. In the dehydration of 1,3-BDO over $Er_2O_3$, however, the formation of UOLs is depressed in $H_2$ flow [45]. The effect of $H_2$ carrier gas depends on the types of reactants and catalysts. Anyway, $H_2$ gas accelerates the formation of 3B1ol in the dehydration of 1,4-BDO over YZrO catalyst, $Y_2Zr_2O_7$. Fluorite $Y_2Zr_2O_7$ has essentially oxygen defect sites on the catalyst surface, as discussed in the following section. It is well known that $ZrO_2$ as well as fluorite $CeO_2$ has oxygen storage capacity [48]. In addition to the original oxygen defects, oxygen vacancies could be generated on the surface of YZrO due to the oxidation of $H_2$. The generation of oxygen vacancies by the oxidation of $H_2$ might be the reason for enhancing the catalytic performance of YZrO in $H_2$ flow.

In Table S2, the variation of the UOLs formation rate with *W/F* was almost constant at conversions lower than 80%: the formation rate was 0.69 mmol m$^{-2}$ h$^{-1}$ at a conversion of 77.6% while that at 51.5% conversion was 0.75 mmol m$^{-2}$ h$^{-1}$. Then, we estimated the effect of external mass transfer using Carberry number (*Ca*) [49]. *Ca* was calculated to be 0.05 at a conversion of 73%. Because an external concentration gradient is absent at *Ca* < 0.05, the effect of external mass transfer would be small at a conversion of lower than 73%. In contrast, at high conversions over 73%, a catalyst efficiency becomes low due to mass transfer limitation in this study. Thus, in Figure 10, some of the formation rate of UOLs calculated from the conversion data over 73% could be underestimated. At high conversions near 100% without 1,4-BDO, subsequent reactions proceeded: the dehydration of 3B1ol to BD and the isomerization of 3B1ol proceed (Figure 7).

Besides, we exhibited that BD was produced as a by-product in the dehydration of 1,4-BDO over YZrO even at 300 °C (Table 6). We have previously reported that 1,4-BDO can be selectively converted to BD via an intermediate of 3B1ol over $Yb_2O_3$ at 360 °C [40]. To establish an efficient catalytic system for the production of BD from 1,4-BDO, we need selective dehydration catalysts for 3B1ol without decomposition to propylene. We could obtain several candidates for the stepwise dehydration of 1,4-BDO (Table 2). Thus, the BD formation in the dehydration of 1,4-BDO will be reported in the following paper in a near future.

The dehydration of alcohols readily proceeds over strong acidic catalysts, but it is not easy that single dehydration of glycols proceeds to produce selectively UOLs [50,51]: the reason is that step-wise reactions of the produced UOLs frequently proceed over acidic catalysts. Catalysts with acid-base properties are reported to be effective for the formation of 3B1ol in the dehydration of 1,4-BDO [33,46]. $NH_3$ and $CO_2$ gases work as weak poisons for $CaO/ZrO_2$ and $Er_2O_3$ catalysts to reduce the yield of 3B1ol whereas the dehydration proceeds even in the $NH_3$ atmosphere. In the dehydration of 2,3-BDO, both $NH_3$ and $CO_2$ also work as the poisons for the formation of 3B2ol over $CaO/ZrO_2$ [52]. In Table 7, $CO_2$ gas works as a poison of basic sites, but gaseous $NH_3$ accelerates the dehydration. Although the reason for the acceleration by $NH_3$ is not clear, an $NH_3$ gas may act as a base catalyst together with catalyst surface for the abstraction of hydrogen from 1,4-BDO. In a similar manner, over pure $ZrO_2$ with acidic property, it is observed that the formation of 3B1ol form 1,4-BDO is accelerated under $NH_3$ flow conditions [33]. As shown in Figure S3, either hydrogen-bonded or weakly adsorbed molecular $NH_3$ acts as a base catalyst. After it removes from the surface during the reaction, $NH_3$ adsorbed on YZrO acted as a poison of acidic sites. In the formation of THF, however, the formation of THF is affected by the acidic sites because the selectivity is depressed by $NH_3$. A similar trend in the THF formation is reported in the dehydration of 1,4-BDO [33,46].

### 3.4. Mechanistic Considerations on the Dehydration of 1,4-BDO over YZrO Catalyst

Although YZrO samples have both acidic and basic sites, the TPD results (Figure 9) cannot explain the results that YZrO samples calcined at high temperatures have high catalytic performance. Thus, it is reasonable that the change in the ratio of exposed crystal surface with large particle sizes could be the main reason for the catalytic performance enhanced in the YZrO calcined at 900 °C.

As shown in Section 3.2 (Table 5), YZrO samples are composed of $Y_2Zr_2O_7$ with an oxygen-defected fluorite structure, which is the same structure of previously reported $Yb_2Zr_2O_7$ [35]. We have discussed structures of active sites for the dehydration of 1,3-BDO using a crystal model of $Yb_2Zr_2O_7$ and proposed probable model structures of active sites. As shown in the previous report [35], well-crystalized $Yb_2Zr_2O_7$ inevitably has oxygen defect sites on the surface. The defect site exposes three cations such as $Zr^{4+}$ and $Yb^{3+}$, which act as an acidic site, and it is surrounded by six $O^{2-}$ anions, which act as a basic site. Because the crystal structure of $Y_2Zr_2O_7$ is the same as that of $Yb_2Zr_2O_7$, cations such as $Zr^{4+}$ and $Y^{3+}$ exposed on the $O^{2-}$ defect site could work as acidic sites, and an $O^{2-}$ anion surrounding the defect site would work as a basic site (adsorption site in Figure 11A). It is reasonable that the selective dehydration of BDOs to UOLs would proceed via an acid-base concerted mechanism over the basic site of $O^{2-}$ and the acidic sites of $Zr^{4+}$ and $Y^{3+}$.

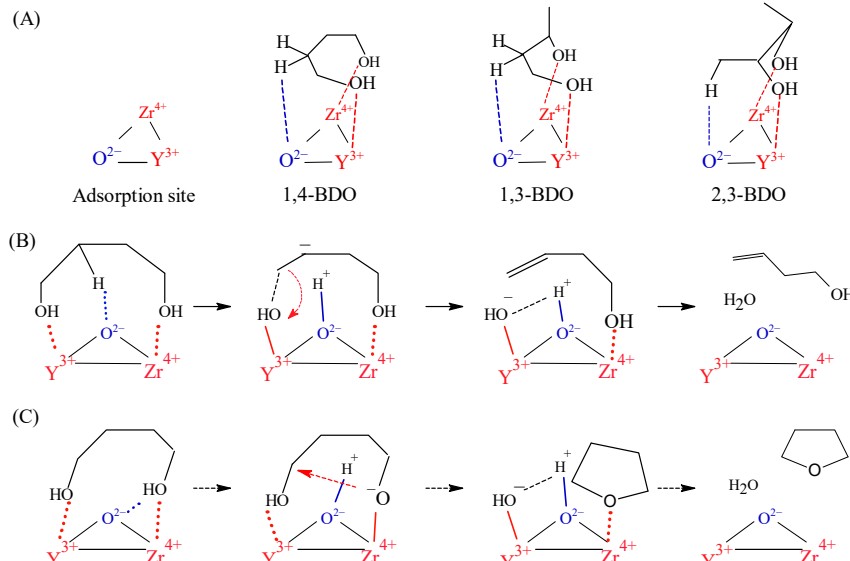

**Figure 11.** Possible coordination structures with adsorption site (**A**) and reaction mechanism for the dehydration of 1,4-BDO to form 3B1ol (**B**) and THF (**C**).

In the dehydration of BDOs, UOLs formation could be initiated by tridentate coordination with two OH groups of BDO on the acidic cation sites and a hydrogen of BDO on the basic $O^{2-}$ site. We speculate possible tridentate coordination structures of BDO adsorbed on YZrO surface (Figure 11A). In Figure 11A, an acceptable adsorption structure of 1,4-BDO has a 6-membered ring, $Y^{3+}$-O-C-C-H-$O^{2-}$, a 7-membered ring, $O^{2-}$-H-C-C-C-O-$Zr^{4+}$, and an 8-membered ring, $Y^{3+}$-O-$C_4$-O-$Zr^{4+}$. In the dehydration of 1,3- and 2,3-BDO, adsorption structures have 7- and 6-membered ring, $Y^{3+}$-O-$C_n$-O-$Zr^{4+}$ ($n$ = 3 and 2), respectively. It is reasonable that the accessibility of hydrogen after the coordination of HO groups, $Y^{3+}$-O-$C_n$-O-$Zr^{4+}$ ($n$ = 3 and 2), determines the reactivity because the anchoring OH group is important for the formation of UOLs [15,34]. The low reactivity of 2,3-BDO is probably caused by large strain in the tridentate coordination structure, which has two 6-membered rings such as $Y^{3+}$-O-$C_2$-O-$Zr^{4+}$ and $O^{2-}$-H-C-C-O-$Y^{3+}$, and a 7-membered ring of $O^{2-}$-H-C-C-C-O-$Zr^4$, as shown in Figure 11A. The 6-membered ring for the fixation of OH groups as $Y^{3+}$-O-C-C-O-$Zr^{4+}$ will induce large strain in the tridentate coordination. On the other hand, in the dehydration of 1,3- and 1,4-BDO, 7- and 8-membered ring coordination, $Y^{3+}$-O-$C_n$-O-$Zr^{4+}$ ($n$ = 3 and 4), respectively, which is looser

than 6-membered ring of $Y^{3+}$-O-$C_2$-O-$Zr^{4+}$ in 2,3-BDO, can allow a hydrogen of BDO to access a basic $O^{2-}$ site to form tridentate coordination structure. It is speculated that the 7-membered ring, $Y^{3+}$-O-$C_3$-O-$Zr^{4+}$, in 1,3-BDO is more stable than the 8-membered ring, $Y^{3+}$-O-$C_4$-O-$Zr^{4+}$, in 1,4-BDO. It is probable that this could be the reason for the difference in the reactivity of BDOs.

Figure 11B proposes a speculative reaction mechanism for the dehydration of 1,4-BDO to produce 3B1ol. The selective dehydration of 1,4-BDO to produce 3B1ol over the defect site is explained by the adsorption as the form of tridentate coordination of 1,4-BDO to the 2 cations and an $O^{2-}$ anion followed by sequential dehydration: the position-2 hydrogen of 1,4-BDO is firstly abstracted by a basic $O^{2-}$ anion and then the position-1 hydroxyl group is subsequently or simultaneously abstracted by an acidic $Y^{3+}$ cation. Probably, the first and the second steps in Figure 11B would proceed simultaneously. Another OH group at position 4 plays an important role in anchoring 1,4-BDO to the catalyst surface. Thus, the selective dehydration of 1,4-BDO to 3B1ol could proceed via the speculative base-acid concerted mechanism, in a similar manner to the 1,4-BDO dehydration catalyzed by $Er_2O_3$ catalyst, which is supported by theoretical calculation [30].

In the formation of THF, a speculative model could be proposed (Figure 11C). The strength of the acidic site is not strong so that carbocation generated by the abstraction of the OH group could not be formed. The TPD results in Figure 9 could explain the result that the YZrO calcined at 600 °C, which has acidic sites stronger than that calcined at 900 °C, shows the higher selectivity to THF than that calcined at 900 °C (Figure 6). Thus, base-acid concerted mechanism can be considered. However, the adsorption structure in Figure 11C is highly strained because it has an 8-membered ring, $Y^{3+}$-O-$C_4$-O-$Zr^{4+}$, and a 9-membered ring, $Y^{3+}$-O-$C_4$-O-H-$O^{2-}$. The suppression of the major by-product, THF, is significant to improve the selectivity to 3B1ol.

## 4. Materials and Methods

### 4.1. Materials

$ZrO(NO_3)_2 \cdot 2H_2O$, $La(NO_3)_3 \cdot 6H_2O$, $Pr(NO_3)_3 \cdot nH_2O$, and $Sm(NO_3)_3 \cdot 6H_2O$ were purchased from Wako Pure Chemicals Industries, Ltd., Osaka, Japan, and $Ce(NO_3)_3 \cdot 6H_2O$ was purchased from Nakalai Tesque, Inc., Kyoto, Japan. Other rare earth (RE) nitrates such as $Y(NO_3)_3 \cdot 6H_2O$ were purchased from Sigma-Aldrich, St. Louis, Missouri, US. In the preparation of $Y_2O_3$-$ZrO_2$ with a Y/Zr ratio of 1, for example, 3.24 g of $Y(NO_3)_3 \cdot 6H_2O$ and 2.26 g of $ZrO(NO_3)_2 \cdot 2H_2O$ were dissolved in 50 $cm^3$ of distilled water. Then, the pH value of the solution was adjusted to 10 with 25 wt.% ammonia water under stirring. After the solution was transferred into 100 $cm^3$ Teflon-lined autoclave, HT treatment was typically performed at 200 °C for 24 h. The recovered precipitate was centrifuged, washed with distilled water, dried at 110 °C for 18 h, and finally calcined at 900 °C for 3 h. Hereafter, the resulted $Y_2O_3$-$ZrO_2$ is named as YZrO. In the case of REZrO, a sample was prepared in the same way as YZrO except using a rare earth nitrate instead of using $Y(NO_3)_3 \cdot 6H_2O$. The resulting zirconate with 50 mol% RE is named as REZrO. For example, the 50 mol% Dy-containing $ZrO_2$ is named as DyZrO.

Reactant chemicals such as 1,4- and 1,3-BDO were purchased from Wako Pure Chemicals Industries, Ltd., Osaka, Japan. 2,3-BDO was purchased from Tokyo Chemical Industry, Co., Ltd. Tokyo, Japan. They were used for the catalytic reaction without any further purification.

### 4.2. Characterization of Catalysts

X-ray fluorescence (XRF) analysis for the estimation of RE content in REZrO samples was performed by EDX-900HS (Shimadzu, Kyoto, Japan). The continuous scanning X-ray diffraction (XRD) was recorded on New D8 ADVANCE (Bruker Japan, Yokohama, Japan) with Cu K$\alpha$ radiation ($\lambda$ = 0.154 nm) to identify the crystal phase of the catalyst. A crystallite size, $D_{XRD}$, of YZrO sample was calculated from the full width at half maximum (FWHM) by Scherrer equation: $D_{XRD} = K\lambda/(\beta\cos\theta)$, where $K$ (=0.9), $\lambda$, $\beta$, and $\theta$ are the shape factor, X-ray wavelength, FWHM, and Bragg's diffraction angle, respectively. The step-scanning XRD of YZrO calcined at 1200 °C was recorded on XRD-7000

(Shimadzu, Kyoto, Japan) with Cu K$\alpha$ radiation in order to refine structure parameter by Rietveld analysis, which was operated by using RIETAN-FP program [53]. High-resolution field emission scanning electron microscope (FE-SEM) image was observed on JSM-7100FA microscope (JEOL, Tokyo, Japan) operated at 15 kV.

The $N_2$ adsorption isotherm of catalyst was measured in a self-made gas adsorption apparatus at −196 °C, and the *SA* of catalyst was calculated from the $N_2$ isotherm fitted with the Brunauer–Emmett–Teller (BET) equation. A particle size of $Y_2Zr_2O_7$, $D_{BET}$, was calculated by the following equation, $D_{BET} = 6/d\, SA$, assuming that the primary particles are spherical or cubic where $d$ is the density of $Y_2Zr_2O_7$ ($d$ = 5.36 g cm$^{-3}$, which is an arithmetic average of 5.03 and 5.68 g cm$^{-3}$ for $Y_2O_3$ and $ZrO_2$, respectively [54]). The calculation of $D_{BET}$ was also performed to compare the morphology of catalyst [55–57].

TPD of adsorbed $NH_3$ and $CO_2$ was performed using BELCATII (Microtrac BEL Corp., Osaka, Japan) with thermal conductivity detector (TCD) to estimate the acidity and the basicity of catalysts, respectively. The measurement conditions are the following: preheating the sample in He flow at 500 °C for 1 h; adsorption of $NH_3$ or $CO_2$ at 100 °C for 1 h; desorption temperature controlled from 100 to 800 °C at an increment rate of 10 °C min$^{-1}$. The TCD detector can detect gases such as water and oxygen desorbed from the catalyst. Thus, we also performed a blank TPD experiment for the sample preheated at 500 °C without adsorption of $NH_3$. Figure S4 shows the blank TCD signal profile as a function of temperature together with the original TCD signals of adsorbed $NH_3$ and $CO_2$. A difference TCD signal between $NH_3$ adsorption and the blank TPD profile was converted to a $NH_3$-TPD profile using molar sensitivity of $NH_3$. In a similar manner, the $CO_2$-TPD profile was obtained from the difference TCD signal between $CO_2$ adsorption and the blank TPD profile. The amounts of acidic and basic sites were estimated from the integral of the desorption profiles of $NH_3$ and $CO_2$, respectively.

### 4.3. Catalytic Reaction

The vapor-phase dehydration of a BDO was performed in a fixed-bed down-flow glass tube reactor under an atmospheric pressure of $N_2$ at a temperature between 300 and 375 °C, typically at 325 °C. After 0.50 g of a catalyst with the granule size of 53–250 μm had been heated in $N_2$ flow at the prescribed reaction temperature for 1 h, a reactant BDO was fed into the reactor at a liquid feed rate of 1.60 g h$^{-1}$ together with an $N_2$ carrier gas at 30 cm$^3$ min$^{-1}$. The reaction effluent at 325 °C was collected at −78 °C every hour, whereas it was collected at 0 °C in the case of the conversion over 80% and the reaction temperature over 325 °C. Differences in the temperature of cold trap were described in the SI file (Table S4). The collected liquid products were identified using a gas chromatograph (GC) equipped with mass-spectrometer (QP5050A, Shimadzu, Kyoto, Japan) and a capillary column (DB-WAX, a length of 30 m, JW Scientific, Osaka, Japan), and they were analyzed by a GC (GC-8A, Shimadzu, Kyoto, Japan) equipped with flame ionization detector and a capillary column of Inert Cap WAX-HT (an inner diameter of 0.53 mm and a length of 30 m, GL-Science, Tokyo, Japan). 1-Hexanol was used as an internal standard for the quantitative GC analysis. The gaseous products, such as BD, propylene, and $CO_2$, collected using a gastight syringe with a volume of 0.5 cm$^3$ were analyzed by another GC-8A equipped with TCD and a packed column (VZ-7, a length of 6 m, GL-Science, Tokyo, Japan) using the $N_2$ carrier gas as an internal standard.

The catalytic reaction was performed during the initial TOS of 5 h to assure the stability of the catalytic activity. All the catalysts showed a little decay in the initial period of 1 h so that we evaluated the catalytic activity we evaluated by averaging the data of conversion and selectivity for 4 h from TOS of 1 to 5 h excluding the first period of 1 h. The mass recovered in the effluent liquid was more than 97% of the mass feed. The error of reproducibility was within 3%. The conversion and selectivity were calculated according to the following equations:

$$\text{Conversion } (\%) = \left(1 - \frac{\text{mole of the recovered reactant}}{\text{mole of the fed reactant}}\right) \times 100 \tag{1}$$

$$\text{Selectivity (mol\%)} = \frac{\text{mole of carbone in the specific product}}{\text{mole of carbon in the fed reactant} \times \text{Conversion}/100} \times 100$$

The poisoning experiment of YZrO catalyst was performed in the catalytic dehydration of 1,4-BDO at 300 and 325 °C to confirm whether acidic and basic carrier gases affect the catalytic activity or not. Instead of using $N_2$ gas, $CO_2$, $NH_3$, or $H_2$ was used as a carrier gas at a flow rate of 30 cm$^3$ min$^{-1}$ in the catalytic reaction. The reaction effluent was collected at 0 °C in the poisoning experiment. In another poisoning experiment, the catalytic reaction was performed in $N_2$ carrier gas flow at 300 °C, after $NH_3$ gas had been flowed at a rate of 30 cm$^3$ min$^{-1}$ and 300 °C for 0.5 h followed by preheating at 300 °C for 1 h.

## 5. Conclusions

The vapor-phase catalytic dehydration of 1,4-BDO to form unsaturated alcohols such as 3B1ol was investigated over 16 REZrO samples together with the dehydration of 1,3- and 2,3-BDO. REZrO with heavy rare earth metals, especially oxygen-defected fluorite $Y_2Zr_2O_7$, showed high conversion of 1,4-BDO to 3B1ol and also high conversion of 1,3-BDO to form 3B2ol and 2B1ol, while 2,3-BDO was less reactive than the other BDOs over REZrO catalysts. The high crystallinity of oxygen-defected fluorite $Y_2Zr_2O_7$ with high surface area was synthesized in the HT treatment of do-precipitate hydroxide in ammonia media. $Y_2Zr_2O_7$ has advantage in the industrial application because of the highest conversion based on weight. The calcination and crystallinity of $Y_2Zr_2O_7$ affected the catalytic activity of the dehydration of BDOs: a calcination temperature of 900 °C or higher was efficient for selective formation of UOLs. In the dehydration of 1,4-BDO, 3B1ol, and THF were the primary products over $Y_2Zr_2O_7$ catalyst, while 3B1ol was further reacted to form 2B1ol and BD. To the best of our knowledge, $Y_2Zr_2O_7$ showed the highest 3B1ol productivity of 27.0 mol h$^{-1}$ kg$^{-1}$ in the dehydration of 1,4-BDO at 325 °C. In the poisoning experiment, both $CO_2$ and $NH_3$ behaved as poisons during the dehydration of 1,4-BDO. This suggests that the reaction proceeds via acid-base concerted mechanism.

**Supplementary Materials:** The following are available online at http://www.mdpi.com/2073-4344/10/12/1392/s1, Figure S1, $N_2$ adsorption-desorption isotherm of YZrO; Figure S2, XRD pattern of YZrO calcined at 900 °C together with PDF data; Figure S3, Changes in catalytic activity of YZrO calcined at 900 °C with TOS with and without $NH_3$ pre-adsorption prior to the reaction; Figure S4, original TCD signal profiles in the TPD measurement of $NH_3$ and $CO_2$ desorbed from YZrO calcined at 600 and 900 °C; Table S1, Dehydration of 1,4-BDO over YZrO calcined at different temperatures; Table S2, Dehydration of 1,4-BDO over YZrO calcined at 900 °C with different space time; Table S3, Dehydration of 1,3-BDO over sixteen REZrO catalysts; Table S4, Dehydration of 1,4-BDO over YZrO calcined at 900 °C in different cold trap.

**Author Contributions:** A.M., Y.M., K.N., F.S. and S.S. summarized the data and prepared the original draft; A.M. and Y.M. prepared samples; A.M., Y.M. and K.N. performed catalytic tests in the dehydration of butanediols; A.M., Y.M., F.S. and R.T. characterized the samples; Y.Y. supervised the analytical work; S.S. designed the project, acquired funding, and edited the manuscript. All authors have read and agreed to the published version of the manuscript.

**Funding:** This work was funded by JSPS KAKENHI KIBAN B, grant number JP18H01784.

**Conflicts of Interest:** The authors declare no conflict of interest.

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
