# Peer review of "Dehydration of Biomass-Derived Butanediols over Rare Earth Zirconate Catalysts"

_catalysts, doi:10.3390/catal10121392_

Round 1

Reviewer 1 Report

The paper entitled “Dehydration of biomass-derived butanediols over rare earth zirconate catalysts" (Manuscript Number: catalysts-999392) is devoted to the preparation, characterization and catalytic activity investigation of rare earth zirconate catalysts for the dehydration of several butanediols. First, sixteen rare earth are investigated and Y-based catalyst was chosen due to its higher catalytic performance. Secondly, the effect of catalyst calcination temperature on the physicochemical properties of the catalyst and on the catalytic performance is evaluated, being determined that a calcination temperature of 900 ºC results in a superior catalytic activity for the studied reaction. Finally, the effect of carrier gas on the catalytic activity for 1,4-butanediol was evaluated for the best catalyst and some mechanistic aspects of the process are addressed. The catalysts presented in this contribution show a fairly good catalytic performance. Characterization and evaluation tests have been designed properly. However, there are some parts of the document that should be improved for publication. A considerable percentage of the references are related to the work carried out by the author’s research group. Language revision by a native-speaker is suggested.

Introduction section should be more elaborated. The importance of biomass for the obtention of butanediols could be highlighted. Most of the introduction is based on the previous work of the group, I suggest to further comment what is done in the literature by other research groups.

Section 2, although meaningful, is hard to follow at some points. I suggest a slight restructuration of the results.

Line 166: DBET is not formally correct. This parameter has been calculated assuming the catalyst to be composed by perfect and non-porous spheres. In general, catalysts are formed by irregular nanoparticle aggregates forming clusters of thousands of microns (or larger), as observed by SEM images in this work. The joining of these nanoparticles, forming large aggregates, causes the textural development (or at least part of it) of the catalyst, which can be estimated by N2 adsorption-desorption. Therefore, Reviewer has concerns on the validity/physical meaning of DBET. In this line, Reviewer wonders about the catalyst particle size used for reaction experiments (This data should be stated in the experimental section).

Line 75- 76: “Zr reagent has Hf as an impurity. This indicates that the REZrO samples have a composition of RE/Zr = ca. 1. LaZrO, PrZrO, CeZrO, and ScZrO samples, however, had a RE/Zr ratio less than 1.” Is Hf also present as an impurity in La, Pr, Ce and Sc-containing samples?

Line 109: “contact time” would be more formally correct if expressed as “space time”.

Lines 163-164: “estimated by using the FWHM of the diffraction peak” should be “estimated using Scherrer equation”.

Table 4: Which peak is being evaluated for FWHM calculation?

Lines 217-218: Figure 6 and 7 appear together, but captions are shown stacked. I would suggest grouping both Figures in one Figure or to separate properly.

Line 238: “…either CO2 or NH3 was adapted as...” Hydrogen was also used for these experiments, so it should be stated there.

Line 275: “property” should be “properties”

Lines 281-282: “Judging from the profiles of NH3-TPD, the acid strength of YZrO calcined at 900 ºC is weaker than that of YZrO calcined at 600 ºC.” The acid strength distribution seems to be not very different for both catalysts, what it is more noticeable is a reduction in the number of total acid sites when rising the calcination temperature. This point should be addressed properly.

Section 3.1, first paragraph (Lines 295-303) could be more elaborated to make the results of the present contribution more attractive to the reader.

Line 304: “in the previous paper” should be “in our previous paper”.

Line 314: Based on which results is the active phase confirmed?

Line 360: “It has been observed” would be more appropriate as “It has been reported in the literature that”.

Lines 370-376: Mass transfer limitations are suggested by authors to explain the catalytic results. The presence or absence of heat/mass transfer limitations should be addressed for the correct interpretation of the catalytic results. On this issue, Reviewer recommends papers such as Catalysis Today 60 (2000) 93–109, Chemical Engineering Journal 378 (2019) 122198 and Chemical Engineering Journal 347 (2018) 741–753.

Line 392: “sure” would be more appropriate as “clear”.

Line 401: “consideration” should be “considerations”.

Reviewer 2 Report

This is a paper about the dehydration of butanediols with a series of rare earth-based zirconia catalysts. The authors did a thorough investigation on the reactivity of all the catalysts by testing different diols (1,3- and 1,4- and 2,3-butanediol) and concluding that the YZrO catalyst is the best, particularly in terms of 3B1ol productivity.

The overall work is sound and well presented. The Discussion session needs extensive rewriting due to poor English language though.

As a general comment, from a literature research, and as well cited, the authors have published a series of articles practically identical to one another in the last years. It appears as though the authors use an always identical mold to cast very similar papers - the differences being, but not always, the catalyst composition and the reactants. This is certainly fair but shows at best a certain lack of understanding of the catalytic system and provides little advancement in the field. For example, the authors have proposed in many of their last publications the very same reaction mechanism without never actually investigating it but just conjuring differences in the atomic radius of the rear earth and different catalytic sites acid sites. To this end, it would be interesting to ascertain which kind of acid and basic site feature the catalysts by FTIR investigations with pyridine and CO2, respectively, and correlate them to the catalytic activity by doing FTIR analyses of some of the reactant adsorbed on the catalysts (or at least on the most active one).

Furthermore, since the morphology of the catalysts seems to be critical to the reactivity, the nitrogen adsorption isotherms should be reported at least in the SI.

Finally, in the present work, the activity of the YbZrO catalyst in the dehydration of 1,3-BDO is reported to be higher than that reported in Molecular Catalysis 473, 2019, 110399 by the same authors. The same for the YZrO for the dehydration of 1,4-BDO in the paper Applied Catalysis A, General 575, 2019, 48-57. Why? It may be due to the different composition and/or preparation procedure? Please clarify.

Reviewer 3 Report

The manuscript is very well-written and I have only few minor comments:

1/ I would like to ask about the carbon balance - this is an important value describing the catalysis. Actually I noted that all selectivities give 100 % in total. This makes perfect sense in theory, however, this is not possible for real measurements. Can you comment on this, please, and correct the values with real carbon balances?

2/ Please round the surface areas to whole numbers. Decimal points do not have any justification considering the error of this method. Can you please show some of the isotherms in supplementary materials and discuss potentially the type of porosity present?

3/ Discussion regarding Fig. 10: Can you also find a connection to the basicity of RE oxides? It seems that  basicity of your materials governs the catalytic activity, right?

4/ Few typos: Line 410 - three should be free?; line 543 - co-precipitate; figure 10 - avarage should be average.

5/ Figure 8, discussion: You describe the gradual decrease of conversion with TOS in the case you use N2 as the carrier gas. Can you mention some numbers in the discussion? The catalyst actually seems to by quite stable...

6/ The mechanism discussed in Fig. 11 is very speculative. Can you support it with some references, experimental observations, or computational details?

Reviewer 4 Report

The manuscript submitted by Sato et al reports on the use of rare-earth zirconate catalyst for the selective dehydration of butanediols to unsaturated alcohols. The authors prepared a large number of catalysts and not only evaluated the influence of the rare-earth element but also of the calcination temperature. The experiments are well described but the discussion needs improvement:
On page 9, the influence of ammonia in the pre-adsorption experiment is interesting but also in contradiction to the NH3-TPD results. In fact, from the latter (Figure 9), one can see that most ammonia will desorb from the catalyst at T<300°C. However, one can imagine that the remaining adsorbed ammonia acts as base catalyst (but will desorb progressively). The authors should reconsider:
- The statement that “NH3 adsorbed on YZrO at 300 °C acted as a poison of acidic sites in N2 flow”
- To perform a control experiment using ammonia co-feeding with respect to their statement “It is possible that either weakly adsorbed NH3 or gas-phase one may act as a catalyst.”
- To perform a control experiment with pre-adsorbed ammonia but employing an MS to detect the desorption of Ammonia during the first 2h
Page 10: The statement “Acidic and basic sites on which NH3 and CO2 adsorbed at temperatures higher than 325°C are considered to be poisoned in the reaction at 325°C.” should be revised. In fact, the adsorption/desorption is a dynamic process which is based on an equilibrium. Hence, if the partial pressure of ammonia or CO2 in gas-phase is low, a significant part of these adsorbed molecules will nevertheless start to desorb. The authors may perform a TPD with a temperature plateau at 325°C in order to verify.
From Figures 6 one can see that for the YZrO catalyst, the selectivity to THF and 3B1ol is independent from the conversion. However, when plotting the data for the different catalysts from table 3, one can see two trends: the selectivity to THF decreases with the conversion whereas the selectivity to 3B1ol increases with the conversion. Hence THF may be an intermediate, which should be considered in the postulated reaction mechanism.

The discussion of the catalytic performance only based on the Ion radius is a bit simple. Catalytic reactions generally guided through redox, acid-base or shape properties of the catalyst. In the current example, it is clear that the acid-base properties are the key-factor. Hence, the authors should perform ammonia and CO2 TPD for all samples and discuss the results on these data instead of the ion radius.
Finally, the mechanistic considerations are not meaningful in the current form. The authors should perform DFT calculation in order to validate the postulated mechanism.
In conclusion, the manuscript does not match the scientific quality for a publication in Catalysis. With respect to the different points that need amelioration, the manuscript should be rejected.

Round 2

Reviewer 1 Report

The paper entitled “Dehydration of biomass-derived butanediols over rare earth zirconate catalysts" (Manuscript Number: catalysts-999392) is devoted to the preparation, characterization and catalytic activity investigation of rare earth zirconate catalysts for the dehydration of several butanediols. I have carefully read the author’s response document and the revised version of the manuscript. The manuscript has been improved. Most of my comments have been adequately sorted out. However, some of them could have been more carefully considered and a more elaborated modification in some aspects of the manuscript could have been accomplished. Language revision by a native-speaker is still suggested.

Introduction section has been just slightly modified. I suggest a more elaborated introduction. Most of the introduction is based on the previous work of the research group. I recommend, if that would be possible, to further include relevant research publications from other research groups.

Line 168: Reviewer has concerns on the validity/physical meaning of DBET. The physical meaning of this doubtful since the assumptions considered for its calculation (perfect and non-porous isolated spheres) are far from real catalyst nature. I suggest reconsidering the use of this parameter. In case you want to use it, please compare with other studies, if available in the literature, dealing with the calculation of this parameter.

Lines 284-288: Reviewer recommend rewriting these lines to make it more clear to the reader. I kindly suggest reshaping it similar to the following sentence: “Acid strength distributions obtained for both catalysts showed little variations in the low temperature range. In the high temperature range (above 400 ºC), the reduction of NH3 released during TPD for the sample calcined at 900 ºC suggest a slight reduction of the stronger acid sites when rising the calcination temperature from 600 to 900 ºC. In addition, a reduction in the total number of acidic sites was clearly appreciated when comparing both YZrO catalysts. Basic strength distribution showed the same trend as the acid strength distribution, when comparing both catalysts.”

Lines 376- 379: Reviewer wonders about why Carberry number and diffusion limitations calculations cannot be shown. If the calculation of these parameters is not done, I would suggest to not going into details with this issue, since no theoretical nor experimental evidence of the occurrence of these phenomena are shown.

Reviewer 2 Report

The authors have answered to some of the points raised during the first round of reviewing and now the paper is suitable for publication. Anyway, it remains the the fact that this paper is practically identical to many others from the very same authors both in its structure and conclusions, regardless of the different type of materials studied. It would be best for the next papers on such an important topic to start doing more in depth characterization, with a focus on the type of acidity involved, and by performing theoretical calculation to support the conclusions on the catalytic pathways.

Author Response

Thank you for the comment. I would like to clarify the type of acidity contributed the catalytic reaction in the following work. We appreciate your time to review our manuscript.

Reviewer 4 Report

The authors have answered to the reviewers comments whereby the scientific quality has increased. Correspondingly the manuscript can be accepted for publication.

Author Response

Thank you for the comment. We really appreciate your time to review our manuscript.

Round 3

Reviewer 1 Report

The paper entitled “Dehydration of biomass-derived butanediols over rare earth zirconate catalysts" (Manuscript Number: catalysts-999392) is devoted to the preparation, characterization and catalytic activity investigation of rare earth zirconate catalysts for the dehydration of several butanediols. I have carefully read the new author’s response document and the new revised version of the manuscript. The manuscript has been improved. My comments have been adequately addressed. Although Reviewer still has some concerns about the use and applicability of DBET, the authors have justified it based on previous studies reported in the literature. The manuscript is suitable for publication in Catalysts.